**communications** engineering

# Architecture for sub-100 ms liquid crystal reconfigurable intelligent surface based on defected delay lines
Robin Neuder ✉, Marc Späth, Martin Schüßler & Alejandro Jiménez-Sáez ✉

Reconfigurable intelligent surfaces, comprised of passive tunable elements, are emerging as an essential device for upcoming millimeter wave and terahertz wireless systems. A fundamental aspect of the device involves the tuning technology used to achieve reconfigurability. Among alternatives such as semiconductors and micro-electromechanical systems, liquid crystal offers advantages including cost- and power-effective large-panel scalability. In this context, conventional liquid crystal-based reconfigurable intelligent surface approaches face limitations in optimizing for bandwidth, response time and loss simultaneously, requiring trade-offs between them. Here we detail an architecture for a liquid crystal-based reconfigurable intelligent surface with compact defected delay lines that provide continuous, 360-degree tunability, enabling fast response time, wide bandwidth and low loss. A reconfigurable intelligent surface with a thin 4.6 $\mu$m liquid crystal layer is designed, fabricated, and characterized, exhibiting response times of 72 milliseconds, insertion losses below 7 dB, and a 6.8 GHz (10.9%) bandwidth at 62 GHz, all while utilizing a lossy glass substrate and gold as a conductor.

Recently, reconfigurable intelligent surfaces (RISs) have gained widespread attention concerning their potential impact in novel millimeter wave (mm-Wave) and terahertz (THz) systems[1-5]. Due to their passive nature, these large tunable reflectors represent a cost- and power-effective approach to overcome challenges in achieving reliable and robust communication links. Two exemplary scenarios for outdoor and indoor communication are depicted in Fig. 1a, b. Despite extensive research already conducted investigating the benefits and limitations of RISs from a signal processing standpoint, only a limited amount of hardware realizations has been published.

From the hardware perspective, a RIS is equal to a tunable reflectarray with an undefined feed. It consists of numerous independently tunable radiating elements forming a passive surface with which an incoming electromagnetic wave can be manipulated, i.e., focused (near-field) or directed (far field) to a user. Via multiple low-frequency control signals (from DC up to a few GHz, depending on the response time), the RIS can generate any possible phase profile on its surface and therefore reflect signals in any directions, acting as a tunable mirror able to rotate and bend arbitrarily.

Hardware realizations can be classified in terms of the tuning technology and the tuning architecture, enabling the reconfigurability of the RIS. In terms of the tuning architecture, RIS realizations can be subdivided into resonant architecture (RA) and delay line architecture (DLA)[6].

### Resonant architecture

In the RA, a tunable material is embedded in between a ground plane and a resonant element, combining phase shifting and radiating properties in one layer. The main advantage of this method is its simplicity, while the design only allows for a low degree of flexibility and is restricted to small bandwidths of few percent, especially for electrically thin substrates.

### Delay line architecture

In the DLA, phase shifting and radiating layer are separated, commonly realized by coupling a radiating element to a dedicated tunable delay line. Although this method requires more design effort, it offers a high degree of flexibility and enables larger bandwidths compared to the RA.

Semiconductor approaches are more widely used for tuning. Particularly with positive intrinsic negative (PIN) and varactor diodes, promising designs have been published[7-9]. Although mostly being located at frequencies below 30 GHz, a recent work showcases a semiconductor-based RIS working at 340 GHz. Additionally, in ref. 10, a highly integrated design based on complementary metal-oxide-semiconductor (CMOS) is presented. These approaches have the main advantage that semiconductors are

Institute of Microwave Engineering and Photonics, Technical University of Darmstadt, Merckstraße 25, Darmstadt 64283 Hessen, Germany.
✉e-mail: robin.neuder@tu-darmstadt.de; alejandro.jimenez_saez@tu-darmstadt.de

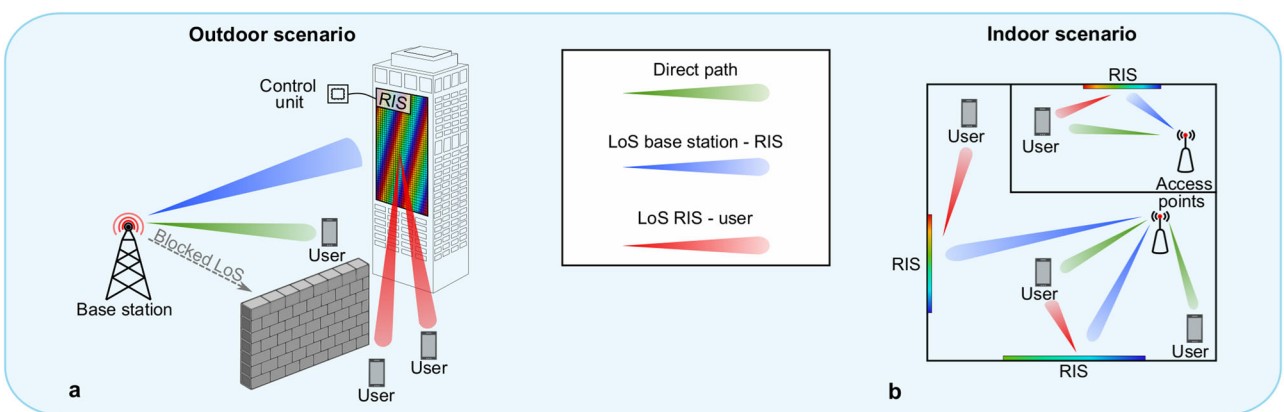

**Fig. 1 | Potential applications for reconfigurable intelligent surfaces (RIS). a** Outdoor and **b** indoor RIS scenarios. LoS Line of sight.

readily available even at very high frequencies. However, since at least one diode is needed per radiating element, cost and power consumption become important issues, as large-scale RISs with hundreds, thousands or even millions of elements are envisioned in order to outperform conventional active relay stations[11]. In addition, these solutions are limited in the number of states, commonly targeting 2–8 (1–3 bit) discrete phase shifts to restrict the number of tuning elements and biasing lines.

An alternative tuning technology to the widely used semiconductors are microelectro-mechanical systems (MEMS)-based switches and mirrors. Here, electrically controlled micro displacements enable phase shifts applicable for RISs at mm-Wave and THz frequencies. MEMS switches are implemented as loaded transmission lines and show similar insertion losses as liquid crystal (LC) delay lines. They have microsecond response times at the cost of a limited number of states (ref. 12 Ch. 4). MEMS mirrors are based on the displacement of a highly reflective surface and therefore show the lowest losses. However, at the moment, the high mass of the mirror and size of the MEMS in two dimensions limit the response time and lead to grating lobes in one steering plane[13]. Finally, although MEMS implementations feature low power consumption, they are not able to provide a low-cost solution for large-scale RISs.

Nematic LC serves as a viable candidate to overcome the issues present in semiconductor and MEMS realizations. With the already existing expertise in large-panel mass production from display technology, its continuous tunability, and the low power consumption, LC presents a promising approach for the realization of cost and power-effective large-scale RISs. Recent publications[14,15] demonstrate a flat-panel phased array with 70000 independently tunable radiating elements and an aperture of 82 cm. LC-RIS shows potential for diverse applications, being considered not only for mm-Wave and THz communication but also for visible light communication[16–18], where extensive knowledge in LC mixtures and devices already exists. However, the implementation of mm-Wave LC-RISs also poses challenges, as LC provides comparably slow response times in the range of milliseconds to seconds, it is temperature sensitive and experiences moderate insertion loss. These obstacles underline the necessity for a sophisticated design of LC-RIS, particularly in the context of choosing the right tuning architecture, i.e. the choice between RA and DLA.

There already exist promising realizations of LC-reflectarrays and LC-RISs based on RA showing the potential for LC-RISs to operate even at THz frequencies. For example,[19] and[20] operating at frequencies around 100 GHz and 400 GHz, respectively. However, one crucial limitation of the RA lies in thick LC layers (10s–100s of μm) needed to achieve sufficient bandwidths, which lead to slow response times (10s of seconds[21]). Lately, efforts emerged to build RA implementations towards millisecond response time[22], but at the expense of considerably decreased LC tunability and increased losses. Hence, with the LC-RA method, trade-offs between bandwidth, loss, and response time are inevitable. In other

words, the optimization of two aspects concurrently requires making concessions in the remaining aspect.

Originally, the DLA was not the ideal choice for LC-RIS, as conventional phase-shifting topologies (e.g., microstrip lines or coplanar waveguides) are not able to provide the necessary characteristics required for achieving full 360° tunability while remaining sufficiently low losses, compactness and fast response times. However, enhancements in LC delay line technology enable new possibilities regarding the DLA, providing the potential to overcome the trade-offs present in the conventional RA. We[23] and Kim et al.[24] proposed a compact, low-loss LC-delay line based on defected ground structures with response times in the range of 10s of milliseconds for LC-RIS.

By application of the DLA with sophisticated defected delay lines, this contribution gives rise to an LC-RIS with the potential of optimizing towards three main aspects, namely bandwidth, loss, and response time simultaneously, without sacrificing one of the aspects, enabling the pursuit of future research directions. For that purpose, a RIS with thin LC layer, namely RIS$_5$ (with $t_{LC}$ = 4.6 μm), is designed and characterized in simulations and measurements, proving the potential of the approach and providing a proof-of-concept. RIS$_5$ demonstrates a response time of 72 milliseconds, maintains insertion losses below 7 dB, and achieves a 6.8 GHz bandwidth (10.9%) at 62 GHz. The contribution is structured as follows: After a detailed specification of the proposed architecture, the unit cell (UC) and its dedicated components are presented. Next, RIS$_5$ is characterized in terms of its beam steering abilities and, finally, we discuss the benefits and limitations of the proposed RIS architecture.

## Results
### Delay line architecture
Figure 2a–c describes the LC tuning mechanism and the architecture of the LC-RIS. As shown in Fig. 2a, LC molecules experience a variable relative permittivity depending on their orientation with respect to the radio frequency (RF) electric field. The orientation of the LC molecules, represented by the director $\vec{n}$, can be controlled by applying a bias voltage between two metal plates, as illustrated in Fig. 2b. Here, an inverted microstrip line (IMSL) delay line is exemplified with a strip and a ground plane as the two conductors. As a result of the dependence of the molecule orientation with regard to the RF electric field, the electrical length of the delay line can be tuned continuously by applying the bias voltage. The DLA is completed by aperture-coupling the tunable delay line to a patch antenna, as depicted in Fig. 2c.

The main difference of the DLA in comparison to the RA is the separation of the phase shifting and the radiating layer, which provides three benefits:

1. As the phase shifting mechanism does not originate from a resonance effect, amplitude variations are minimized and the bandwidth is mainly dictated by the radiating element.

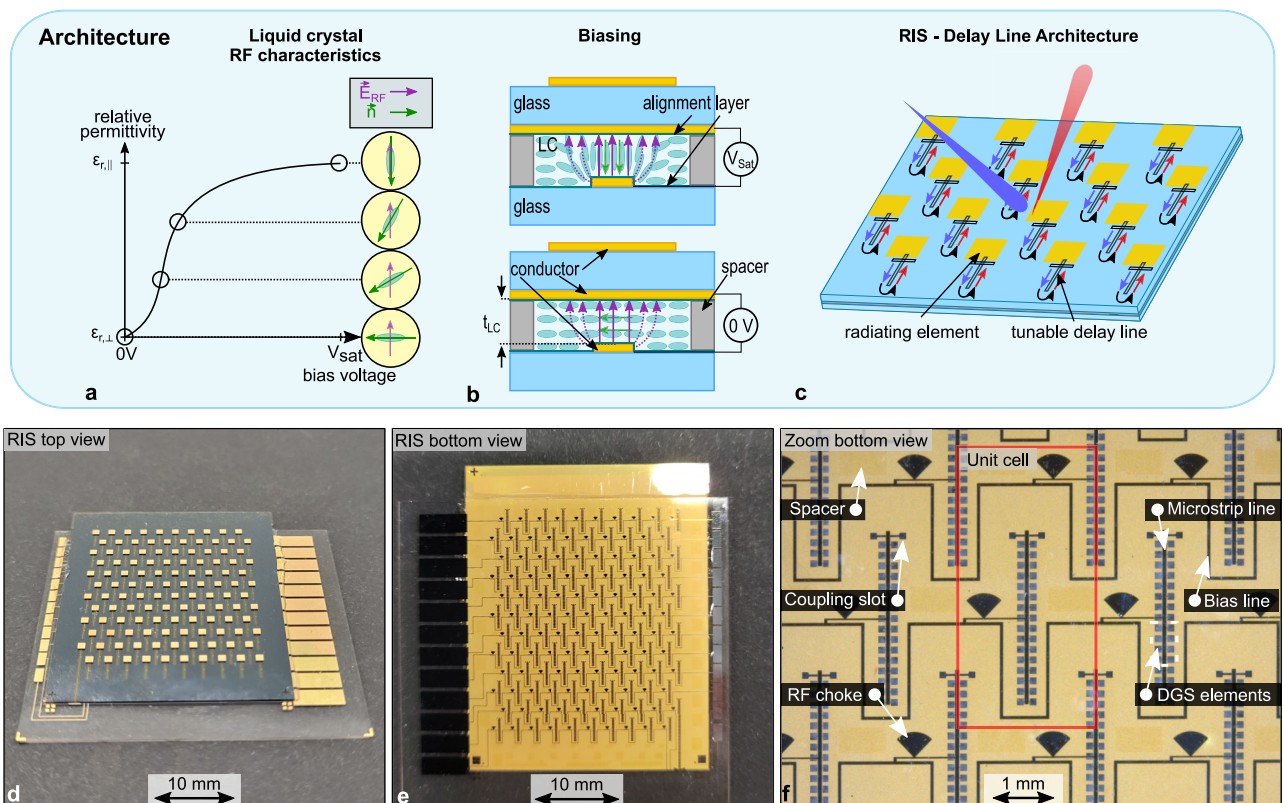

**Fig. 2 | Architecture of the LC-RIS. a** LC-molecule alignment (represented by director $\vec{n}$) with respect to the RF electric field and the resulting relative permittivity in relation to an applied bias voltage. **b** LC-filled IMSL in the fully biased (Top) and unbiased (Bottom) case. **c** RIS layout from top view, with the blue arrows indicating the incident wavefront and the forward traveling wave in the tunable delay line. Black arrows indicate the reflection at the open end of the delay line. Finally, red arrows indicate the backward-traveling wave inside the tunable delay line and the reflected wave. **d** Top view and **e** bottom view of fabricated RIS$_5$. **f** Zoom in on bottom view of RIS$_5$. The red box delimits the unit cell. RF Radio frequency, LC Liquid crystal, $t_{LC}$ Liquid crystal layer thickness, RIS Reconfigurable intelligent surface, V$_{sat}$ Saturation voltage, IMSL Inverted microstrip line, DGS Defective ground structure.

2. The radiating layer can be realized with an electrically thick dielectric material to maximize bandwidth, while the LC layer ($t_{LC}$) can be kept thin. A thin $t_{LC}$ is decisive, as the switch-on response time $\tau_{on}$ (from $\varepsilon_{r,\perp}$ to $\varepsilon_{r,\parallel}$) and switch-off response time $\tau_{off}$ (from $\varepsilon_{r,\parallel}$ to $\varepsilon_{r,\perp}$) increase quadratically with the LC thickness, $t_{LC}$[25].

3. The DLA provides high flexibility, as the tunable delay line can be engineered towards application-specific requirements with relaxed constraints of the selected radiating layer topology.

With the aforementioned benefits, it is possible to optimize a RIS towards low loss, wide bandwidth, and fast response time simultaneously. In light of the above, the design of a sophisticated tunable delay line is crucial for the performance of the DLA LC-RIS. Depending on the application, the tunable delay line should satisfy four conditions[26]:

i. High figure of merit (FoM), i.e. low losses
ii. Low normalized length (NL), i.e. high compactness
iii. Fast response time
iv. Constant group delay

The fourth aspect should be fulfilled to avoid extensive differences in differential phase shifts over bandwidth, which would cause undesired beam-squinting effects. As a conventional IMSL does not meet (ii), and can only meet either (i) or (iii), we recently proposed a more sophisticated delay line topology, the defected ground structure inverted microstrip line (DGS-IMSL)[23].

Besides the numerous benefits of the DLA compared to the RA, there exist some minor trade-offs: The finite length of the delay lines restricts the option of choosing arbitrarily small element spacings, yet it still allows for achieving spacings that prevent grating lobes. In addition, the DLA requires a finer manufacturing resolution than the RA (in this work, the smallest details have a size of 20 μm).

This contribution presents the design, fabrication, and characterization of an LC-RIS with $t_{LC} = 4.6$ μm, RIS$_5$. The radiating elements, which consist of patch antennas aperture coupled to the DGS-IMSL delay lines, are arranged in a triangular pattern as shown in Fig. 2d, e. The unit cell is displayed in Fig. 2f.

Table 1 provides an overview of the utilized materials. In order to reduce the fabrication effort, the biasing lines, which set the bias voltage of the individual delay lines, are decoupled from the RF signal by $\lambda/4$-stubs. Furthermore, the RIS is biased in columns, which results in 1D beam steering capabilities.

### Unit cell characterization

From the RF perspective, a RIS is a periodic repetition of identical elements, the so-called UC. For the proposed DLA, a UC comprises a tunable delay line and a radiating element, defining the capabilities of the LC-RIS. Figure 3 depicts the simulation and measurement results of a UC with $t_{LC} = 10$ μm.

A first important performance metric is the loss of the tunable delay line. With the DGS-IMSL approach, a thin LC layer can be realized with low loss compared to a conventional IMSL, as conductor loss becomes independent of layer thickness. For example, a DGS-IMSL with $t_{LC} = 4.6$ μm can be realized with a fraction of only 38% loss compared to a $t_{LC} = 20$ μm IMSL delay line[26]. The fabricated DGS-IMSL delay line and the on-wafer measurement setup are illustrated in Fig. 3a, b. The measured S-Parameters are demonstrated in Fig. 3d. As exemplified in Fig. 3f, the presented tunable delay line achieves a high FoM (see "Methods" section) up to 81°/dB

**Table 1 | Materials used for the fabrication of RIS$_5$**

| Specification | Material | Thickness (μm) | Permittivity | tan δ | Reference |
|---|---|---|---|---|---|
| Conductor | Gold | 1.5 | – | – | – |
| Glass | AF32 | 300 | 5 | 0.011 | 34 |
| Liquid crystal | GT7-29001 | 4.6 | 2.45–3.53 | 0.0116–0.0064 | 35 |
| Spacer | SU8-3005/-2007 | 6.1 | 3.25 | 0.027 | 36 |
| Alignment layer | Nylon-6 | ≈0.05–0.1 | 2.99 | 0.008 | 37 |

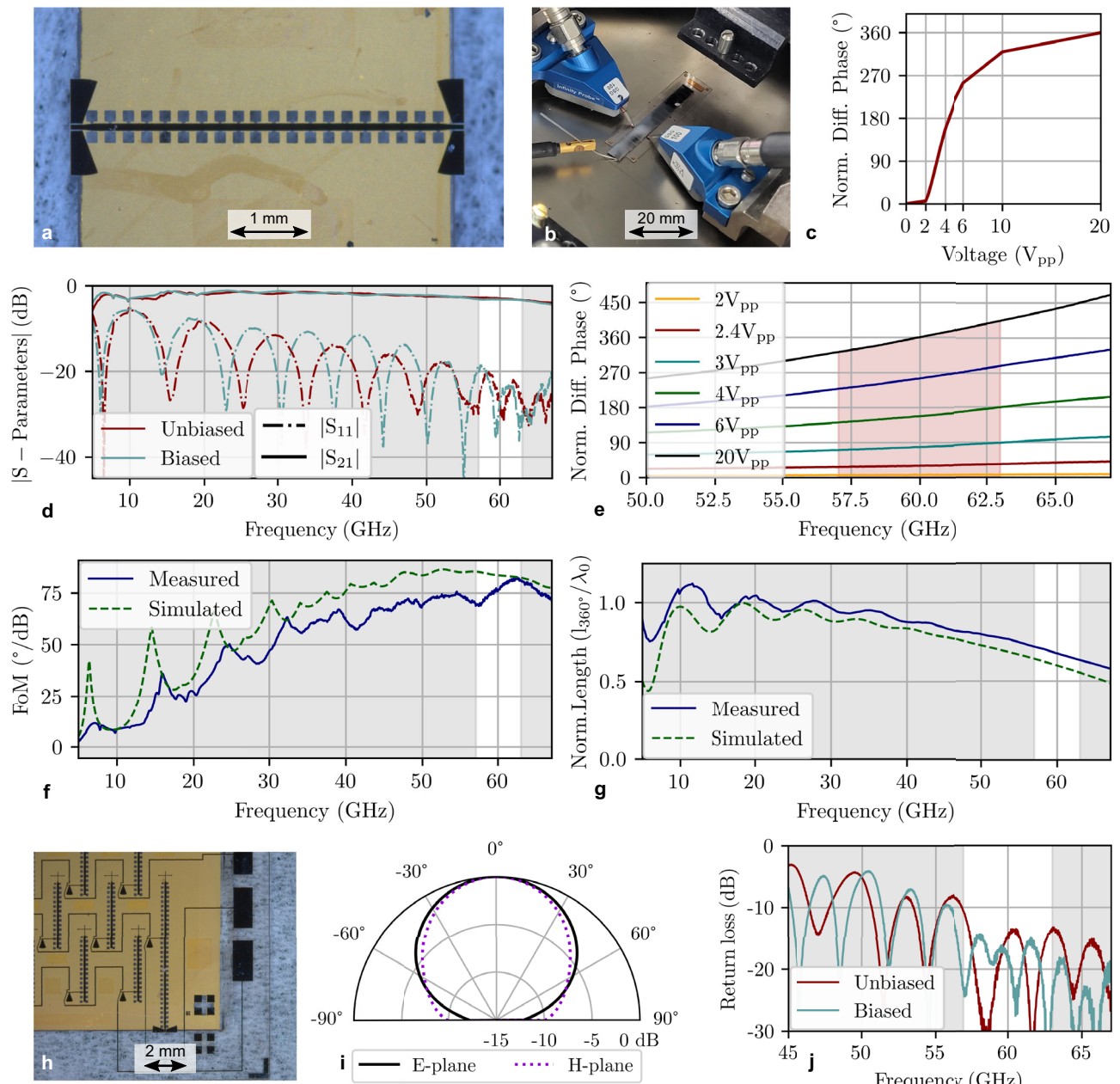

**Fig. 3 | Unit cell characterization for an LC-RIS based on defected delay lines.**
**a** Picture of characterized tunable delay line, including DGS-IMSL to on-wafer transitions. **b** Picture of on-wafer measurement. **c** Phase response over bias voltage. **d** Measured S-Parameters of the DGS-IMSL delay line in the unbiased and biased case. **e** Measured differential phase in the tunable delay line for increasing bias voltages. The red area marks all achievable differential phases. **f** Measured and simulated FoM. **g** Measured and simulated NL. **h** Picture of RIS edge element utilized for on-wafer antenna return loss measurements. **i** Simulated radiation pattern of the unit cell. **j** Measured return loss of the radiation element. The gray and red shadings in (**d**–**g**, **j**) highlight frequencies from 57 to 63 GHz. LC Liquid crystal, RIS Reconfigurable intelligent surface, V$_{sat}$ Saturation voltage, IMSL Inverted microstrip line, DGS Defective ground structure, FoM Figure of Merit, NL Normalized length.

**Table 2 | Comparison of RIS$_5$ and RIS$_{10}$ at 60 GHz**

| Realization | $t_{LC}$ (µm) | NL ($\lambda_0$) | FoM (°/dB) | UC Loss (dB) | $\tau_{on}$ (ms) | $\tau_{off}$ (ms) |
|---|---|---|---|---|---|---|
| RIS$_{10}$ | 10 | *0.6*/**0.67** | 83/78 | *7.4* | *70* | *340* |
| RIS$_5$ | 4.6 | *0.46* | 84 | 6.6 | **15** | **72** |

Bold values are measured, while italic values are either simulated or estimated.

$t_{LC}$ liquid crystal layer thickness, *FoM* figure of merit; i.e. delay line loss, *NL* normalized length; i.e. delay line compactness, *UC* unit cell, $\tau_{on}$ *and* $\tau_{off}$ switch-on/off response times, *RIS* reconfigurable intelligent surface.

**Table 3 | Simulated loss in RIS$_5$**

| Source of loss | Delay line | Radiating element | Unit cell |
|---|---|---|---|
| Conductor | 2.44 dB | 0.62 dB | 3.06 dB |
| Dielectric | 2.44 dB | 1.1 dB | 3.54 dB |
| Liquid crystal | 1.89 dB | 0.39 dB | 2.27 dB |
| Glass | 0.55 dB | 0.72 dB | 1.27 dB |
| Total | 4.88 dB | 1.72 dB | **6.6 dB** |

Liquid crystal and glass losses add up to dielectric loss. Dielectric and conductor loss add up to total loss.
The bold value refers to the total power lost in a unit cell.

between 57 and 63 GHz. This corresponds to a maximum loss of 4.4 dB for full 360° tunability.

A second important measure of the tunable delay line is its compactness. For this purpose, the NL is defined (see "Methods" section). The triangular grid configuration allows for considerable separations between radiating elements, offering substantial space for the delay line (2× element spacing), particularly for linearly polarized RIS. Still, a comparatively compact delay line is advantageous to avoid unnecessary space consumption and meandering. The measured NL corresponds to 0.65 $\lambda_0$ as shown in Fig. 3g.

A merit of LC is its continuous tunability, providing a continuous phase shift controllable by the applied bias voltage. Figure 3c, e depict the differential phase shift in terms of the applied voltage and over-frequency, respectively. Due to the thin LC layer, only limited voltages up to 20 $V_{PP}$ are required for full 360° tunability. This is an additional benefit compared to the RA in which thicker LC layers are needed for wide bandwidth. The results in Fig. 3e show that differential phase shifts are not highly non-linear over frequency, allowing for wide bandwidth applications not limited by beam-squinting.

The simulated E-plane and H-plane radiation patterns of the unit cell are featured in Fig. 3i. In addition, Fig. 3j shows the measured return loss of the antenna element on an edge element of the LC-RIS (as displayed in Fig. 3h). The results reveal good matching below −10 dB for frequencies above 57 GHz for both extreme states of the LC. The small cross-sensitivity of the radiation element to the LC state is another benefit of the DLA. In contrast to the RA, the electrical size of the patch antenna is barely altered with the change of the LC molecules' orientation, since the radiating and phase-shifting layers are sufficiently isolated from one another.

The good agreement between simulation and measurements of the UC verifies the reliability of the full-wave simulations in the design process, as well as the adequate manufacturing tolerances. Such reliability is key for the realization of real RIS and its practical use since the acquisition of data about the RF performance of the single elements is not possible without intensive characterization efforts.

Two RISs have been designed and characterized, RIS$_{10}$ and RIS$_5$, with $t_{LC}$ = 10 µm and $t_{LC}$ = 4.6 µm, respectively. Simulation results of the delay line for RIS$_5$ are presented in Supplementary Fig. 1. Due to the similar characterization and results, this contribution presents the results of the RIS$_5$ implementation, achieving the most remarkable performance results in terms of response time, bandwidth, and loss. For completeness, both RISs are compared in Table 2. Pictures and bistatic measurement results of RIS$_{10}$ are provided in Supplementary Figs. 3 and 5.

The radiation element adds further loss to the UC in addition to the already specified delay line loss. All loss contributions for RIS$_5$, specified at 60 GHz, can be found in Table 3. For the fabricated RISs, the tunable delay line is designed to be longer than the required length predicted by simulations to assure full 360-degree differential phase shifts also in the presence of fabrication tolerances. The longer the length, the larger the loss, resulting in slightly higher losses. Taking this into account, total UC loss could be further reduced to 6 dB. In addition, if the lossy glass AF32 and gold would be replaced with quartz glass (tan δ set to 0.0004) and copper, respectively, simulations show that the UC loss could be reduced to 4.6 dB.

## Bistatic measurements

The RIS$_5$ consists of an array of 12 × 10 UC elements arranged in a triangular pattern. In Fig. 4a, the measured response time of an independent 4.6 µm thin delay line sample is shown. $\tau_{on}$ corresponds to approximately 15 ms, while $\tau_{off}$ is 72 ms, demonstrating the feasibility of LC-RIS with sub-100 millisecond response times. In Fig. 4(b), the measured radiation diagram of RIS$_5$ is given, showing particularly good beamsteering capabilities from −50° to +50°. The undesired received power around 0° mainly arises from environmental reflections, which could not be completely isolated from the desired response of the RIS (see "Methods" section). Still, the response of RIS$_5$ is clearly visible for all desired steering angles. In Fig. 4c, heat maps are given, illustrating the response of the RIS over both, azimuth angle and frequency. RIS$_5$ accomplishes a 6.8 GHz (10.9%) −6 dB bandwidth at 62 GHz for angles between −30° and 30°. In addition, despite the dispersive group delay of the low-pass delay line[23], the RIS experiences imperceptible beam-squinting effects. Table 4 summarizes the characteristics of RIS$_5$.

## Discussion

RISs are viable tools to support mm-Wave and THz wireless communication and sensing systems. Large RISs with thousands or millions of elements are needed, with side lengths of even several meters, so that cost and power-efficient RISs will play an important role. Furthermore, losses should be as low as possible, a sufficient bandwidth must be reached, response times must be fast enough to adapt to a dynamic environment, and wide-angle steering capabilities must be provided. This section discusses the suitability of the proposed DLA LC-RIS to fulfill these requirements. The discussion is divided in limitations of the presented results and the benefits of the architecture and concludes with a suggestion of potential applications.

### Limitations

#### Biasing

Instead of λ/4-stubs, Indium Tin Oxide (ITO)[27] could be used for biasing as in liquid crystal displays (LCDs), reducing the required space for the biasing circuitry. 2D beam steering capabilities can be achieved with the employment of thin film transistors. With these, element-wise biasing can be accomplished in a matrix, meaning that only N (number of rows) + M (number of columns) bias lines are needed[27].

#### Polarization

In this contribution, a single polarized RIS is presented. Nonetheless, the proposed architecture also enables dual-polarized RISs, for example by making use of other delay line topologies, such as bandpass-based delay

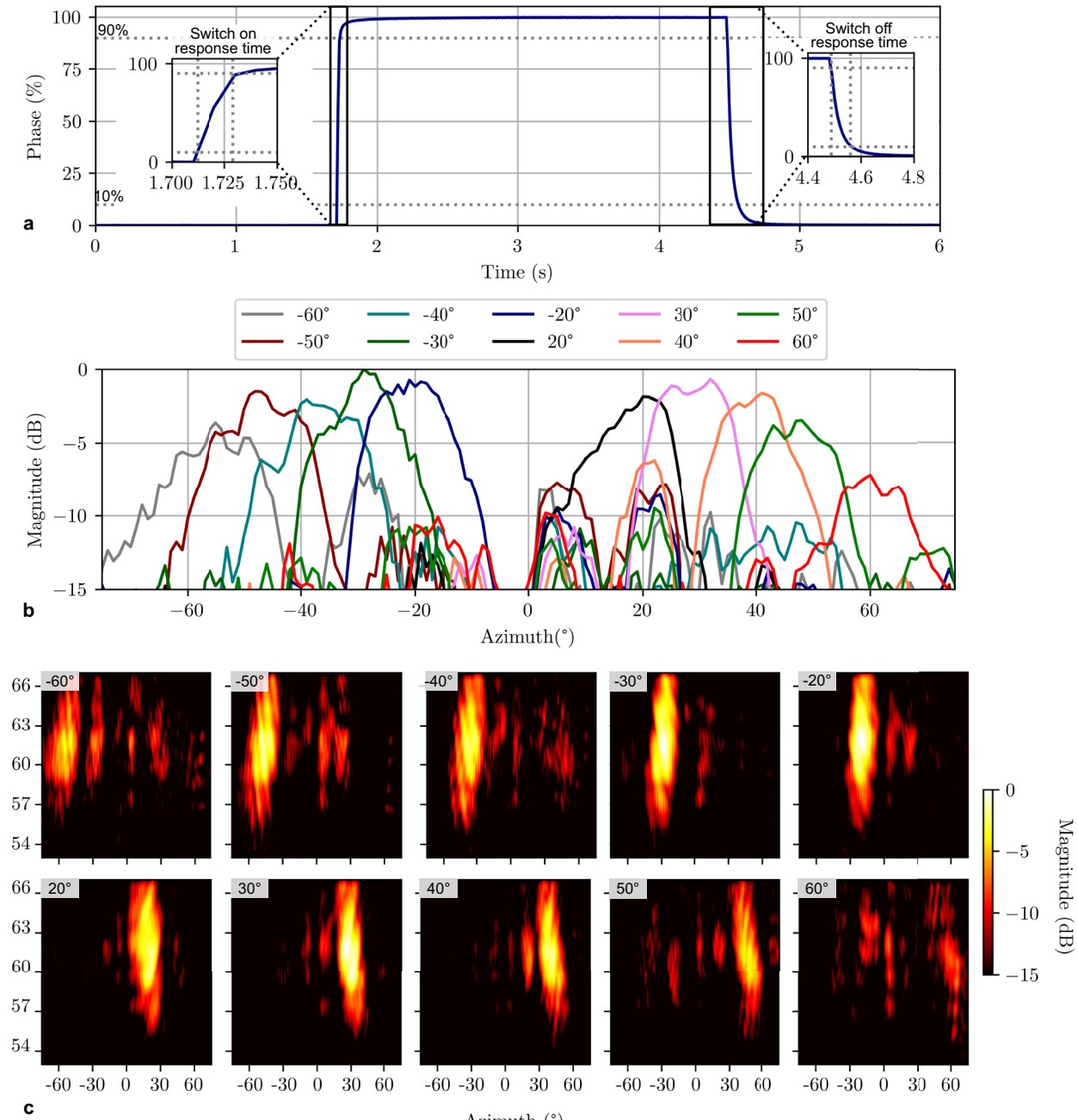

**Fig. 4 | Characterization of RIS₅. a** Measured switch-on ($\tau_{\text{on}}$) and switch-off ($\tau_{\text{off}}$) response time of a liquid crystal delay line with a liquid crystal thickness $t_{\text{LC}} = 4.6\,\mu\text{m}$. **b** Measured normalized response of RIS₅ over azimuth at a frequency of 61 GHz for different voltage configurations. **c** Heat maps, illustrating the measured normalized broadband response of the reconfigurable intelligent surface (RIS) for desired steering angles between −60° to −20°, and 20°–60°, respectively.

lines[26]. In this context, one radiating element can be coupled to two independent tunable delay lines, one for each polarization. Hence, although the mechanism for biasing the LC is the same as for the RA, the dual-polarized DLA additionally enables independent polarization control within a single radiating element, representing another benefit in comparison to the RA. For the latter, at least two radiating elements are required for separate control of both linear polarizations[28].

Limitations of the presented RIS are mainly technological and, thus, solvable by techniques already applied in commercial LCD manufacturing processes. However, temperature dependence of LC, which was not analyzed in the scope of this contribution, and response times are also limitations related to LC-RISs in general. Temperature can be monitored with temperature sensing systems. With this information, the biasing of the delay

lines (i.e. the phase shift) can be adjusted towards optimal performance for a specific temperature. The response time can be reduced considerably with the DLA in comparison to LC-RISs based on the RA as stated in the following section.

## Benefits
### Cost
Compared to semiconductor-based RISs, for which each UC requires at least one diode, LC-RISs can be built cost-effective. With the existing expertise in LCD fabrication, large-scale RISs can be manufactured efficiently in large quantities. In comparison to RA LC-RISs, the DLA enables LC layers with thickness of only few μm. Thus, cost of LC can be considerably reduced compared to thick LC layers[6]. For example, the LC

**Table 4 | Characteristics of RIS$_5$**

| $t_{LC}$ (µm) | Elements | Spacing ($\lambda_0$) | NL ($\lambda_0$) | FoM (°/dB) |
|---|---|---|---|---|
| 4.6 | 120 (12 × 10) | 0.45 | *0.46* | *84* |
| UC Loss (dB) | $\tau_{on}$ (ms) | $\tau_{off}$ (ms) | Bandwidth (%)$^a$ | Beam steering angles |
| *6.6* | **15** | **72** | **10.9** | **−50° to 50°** |

Bold values are measured, while italic values are either simulated or estimated. Values are provided for 60 GHz.

$t_{LC}$ liquid crystal layer thickness, *FoM* figure of merit; i.e. delay line loss, *NL* normalized length; i.e. delay line compactness, *UC* unit cell, $\tau_{on}$ and $\tau_{off}$ switch-on/off response times.

$^a$−6 dB overlapping bandwidth for angles between −30° and 30° (See Supplementary Fig. 2).

amount needed for a 5 µm RIS would be 20 times smaller than for a 100 µm LC layer, a common thickness for RA LC-RIS.

### Power consumption

Biasing of the LC molecules is achieved by applying a bias voltage between two metal plates filled with an insulator (LC). Hence, only minor leakage currents flow, leading to low power consumption[25].

### Beam steering capabilities

Wide-angle beam steering capabilities from −50° to +50° are presented. Further improvements could be achieved with decreased element spacing and the application of wide-angle impedance matching layers. In addition, due to its continuous tunability, LC enables more accurate beams than discrete approaches such as PIN diodes.

Low cost and -power consumption are benefits featured in almost all LC-RIS implementations. An important scope of this paper, however, was to combine these benefits in an LC-RIS with the potential of optimizing for bandwidth, response time, and loss simultaneously, which is an attribute missing for conventional LC-RIS based on RA.

### Bandwidth

The DLA LC-RIS is able to provide broadband performance, which is mainly determined by the bandwidth of the radiating element. In particular, the possibility to design tunable delay lines with nearly constant group delay allows for broad bandwidths with stable beam steering angle. The achieved bandwidth (−6 dB bandwidth of 10.9% at 62 GHz) could be further increased by means of an optimized radiation element and/or by using a thicker radiating substrate with lower permittivity. In addition, the element spacing can be reduced for further improvement[29]. For example, a reflectarray with an aperture-coupled microstrip line-fed patch antenna can achieve 3 dB gain bandwidths of 16.5%[30].

### Response time

Response time is one of the most critical aspects concerning LC-RIS. In comparison to RA LC-RIS implementations, for which response times in the range of few seconds ($\tau_{on}$) and 10s of seconds ($\tau_{off}$) are expected, the presented DLA implementation can provide $\tau_{on}$ and $\tau_{off}$ corresponding to 15 ms and 72 ms, respectively. Hence, this contribution gives rise for LC-RISs with response times in the lower ms range. $\tau_{off}$ could be reduced by further reducing the layer thickness, i.e., <2 ms for $t_{LC} = 1$ µm and by applying overdriving and undershooting techniques already applied in LCD applications[31]. In addition, polymerizable monomers dispersed in LC mixtures increase anchoring forces and hence, reduce the response time[32]. However, this comes at the cost of higher losses and the requirement for higher voltages.

### Loss

Loss in the UC has been characterized to 6.6 dB. Also, it has been shown that by choosing an appropriate delay line topology, loss becomes almost independent of LC layer thickness. Hence, due to the separation of radiating and phase-shifting layer, DLA implementations can provide comparable losses to RA implementations with comparably thick LC layers without sacrificing slow response times or bandwidth. Losses could be further reduced either by improving the FoM of the tunable delay line and/or by replacing the glass substrate with a low-loss glass down to 4.6 dB considering the given architecture.

### Application perspective

Since LC-RISs exhibit a comparably slow response time, their applicability in communication scenarios is constrained. Nonetheless, they possess significant potential for current 5G or upcoming 6G networks due to the aforementioned benefits. One potential application is the illumination of areas that lack sufficient coverage. For example, considering the near-transparent characteristics of LC, LC-RISs may find deployment in window facades of buildings in cities. This utilization would facilitate the reflection of RF waves towards specified directions and/or their transmission into the building, augmenting received power levels. Additionally, envisioned use cases involve the dynamic alignment of the RIS beam to track moving objects.

Given the modest response time prerequisites in illumination scenarios, LC-RISs based on the DLA hold promise for efficient operation. In tracking scenarios, the required response time depends on the velocity and distance of the tracked object. Bandwidth requirements vary depending on the use case. For instance, the 5G band n257 (26.5 GHz and 29.5 GHz) demands a 10.7% bandwidth, which could already be met by the proposed RIS. Yet, the European wireless network standard IEEE 802.11ad (57 GHz–66 GHz) allocates a 14.6% bandwidth, suggesting that further research is needed to widen the bandwidth for the complete band to be covered. Nonetheless, the LC-RIS's ability to be scaled towards desired operating frequencies (upwards and downwards) remains a benefit for its application in future communication networks. Since the loss introduced by the LC-RIS only depends on the UC loss and not on its size, it can be counteracted by enlarging the dimensions of the RIS. Beyond the aforementioned communication applications, LC-RISs also present prospective utility in sensing systems.

### Conclusion

This paper describes a new architecture for reconfigurable intelligent surfaces based on defected LC-delay lines. In contrast to RIS relying on the detuning of resonant elements in the unit cells, in this approach, the phase shifting for the backscattered signals is achieved by tunable delay lines for each antenna element. The architecture allows for future LC-RIS which combine wide bandwidths, fast response times, and moderate loss while featuring a physically cost-effective large-area surface since the technical implementation originates from display technology. The first realized LC-RIS prototype exhibits 72 ms response time, insertion losses of 6.6 dB, and a 10.9% bandwidth. While the feasibility of the delay line architecture was presented, further component optimization, technological improvements, and low-loss materials will allow for additional enhancements, including reduced losses (below 5 dB), faster response times (few milliseconds), increased bandwidths (16.5%[30]), the capability for element-wise biasing and dual-polarized RISs. Overall, the proposed implementation provides a promising alternative approach for LC-RISs by addressing key challenges. It initiates a new research direction and opens new possibilities for the realization of RISs for future mm-Wave and THz communication and sensing systems.

### Methods
#### Measurements

All measurements are conducted with a PNA-X N5247A from Keysight Technologies. The on-wafer measurements are executed with I67-GSG-100 probes. Bias voltages are applied with a 1 kHz square wave signal. For the bistatic measurement of RIS$_5$, bias voltages are set with a DAC60096 EVM from Texas Instruments, controlled with a Python script. The DAC provides a 12-bit output voltage range between −10.5 V and 10.5 V, so that the LC can be biased almost continuously. The bias voltage is applied onto bias

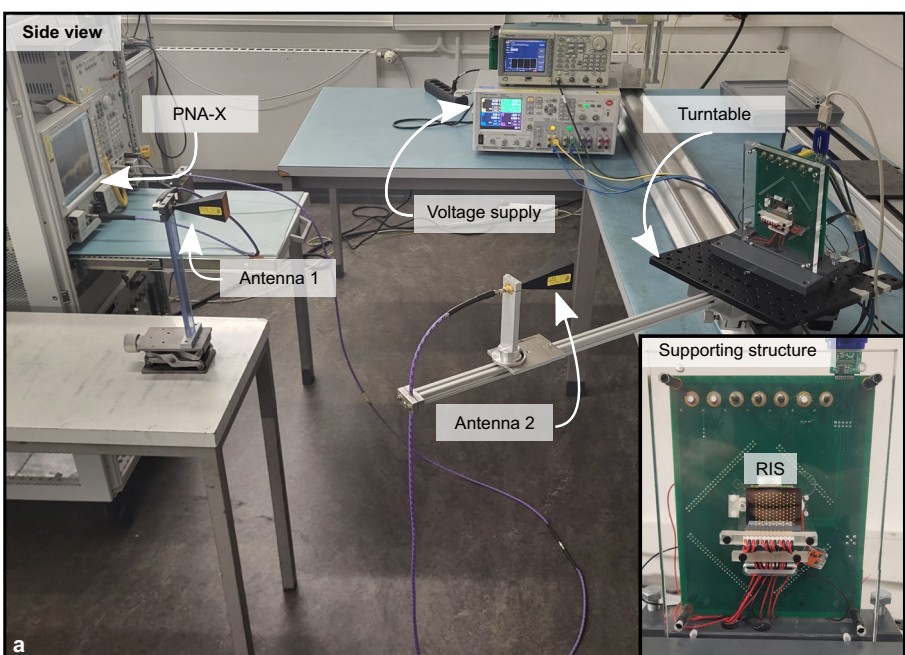
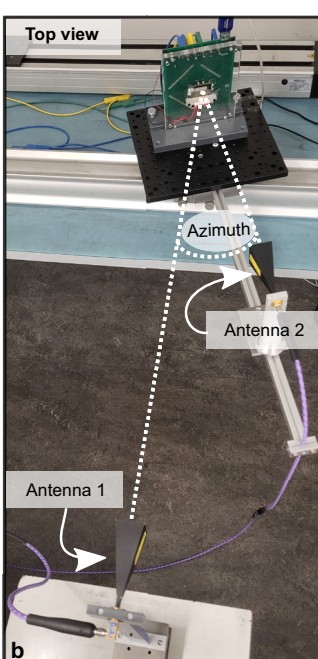

**Fig. 5 | Measurement setup for bistatic measurements. a** Side view of measurement setup with a picture of the RIS embedded in its supporting structure and **b** top view of measurement setup. RIS (Reconfigurable intelligent surface), PNA-X (Keysight Technologies network analyzer).

electrodes on the glass samples and connected to the microstrip lines via gold bias lines. Since the RIS is biased only in columns, 12 bias voltages are applied at the same time to achieve the desired phase shift. In addition, the ground is set by an additional pin connected to the ground plane. Bistatic measurements are conducted with two MI-WAVE V-band horn antennas with 25 dBi gain pointing towards the RIS. The bistatic measurement setup is illustrated in Fig. 5a, b. Having a slightly larger aperture than the RIS, the far field of the horn antenna lies at 90 cm.

Antenna 1 and antenna 2 are located at a distance of 1 m and 55 cm from the RIS. Antenna 2 co-rotates with the RIS and therefore remains in frontal incidence of the antenna for all measurements. The RIS is supported by a 20 cm high, 15 cm wide, and 1 cm thick Plexiglas. Due to the large aperture of the supporting structure with regard the RIS ($\approx$2.8 cm $\times$ 3.3 cm), two reference measurements are performed to isolate the desired response of the RIS from its environment. In addition, $RIS_5$ has a preset phase shift towards 15° in elevation, which facilitates the isolation of the desired response of the RIS from the cluttered environment. Two reference measurements are performed in order to isolate the desired measurement signal from the undesired influence of its supporting structure. To separate static and dynamic influences, one reference measurement is conducted with all delay lines set to 0° (0 $V_{PP}$) and a second with all delay lines set to 180° ($\approx$3.4 $V_{PP}$). By applying the following mathematical operation:

$$S_{21,\text{RIS}} = S_{21,\text{Meas}} - 1/2 \cdot (S_{21,\text{Ref } 0°} + S_{21,\text{Ref } 180°}), \tag{1}$$

theoretically, all undesired influences are filtered out and only the desired response of the RIS remains. Note that phase information is required for reference measurements. To refine the isolation of the desired response of the RIS from environmental influences, a time gating window of 1.5 ns duration is applied between 4.5 ns and 6 ns.

### Performance metrics
For the evaluation of the tunable delay lines, two metrics are introduced. The FoM, which defines the losses of the tunable delay line, and the NL, which provides a measure for the compactness of the

tunable delay line. The FoM is defined as

$$FoM = \frac{\Delta\phi_{\max}}{IL_{\max}}, \tag{2}$$

with $\Delta\phi_{\max}$ and $IL_{\max}$ corresponding to the maximum differential phase shift and maximum insertion loss of the tunable delay line. The NL is defined as

$$\text{NL} = \frac{1}{\lambda_0} \frac{360° \cdot l_{\text{phys}}}{\Delta\phi_{\max}}, \tag{3}$$

with $\lambda_0$ being the free-space wavelength, $l_{\text{phys}}$ the physical length of the tunable delay line and $\Delta\phi_{\max}$ the maximum phase shift. Hence, NL is defined as the length of the tunable delay line needed for full 360° tunability with respect to the free-space wavelength $\lambda_0$, i.e., the delay line length at least required in the final RIS design.

### Simulation
Electromagnetic simulations have been conducted with CST Studio Suite. The DGS-IMSL has been thoroughly optimized for the operation around a center frequency of 60 GHz. For this purpose, the position and dimension of its defects as well as the width of the microstrip line are optimized. While the dumbbell-shaped defects add a series inductance to a conventional IMSL, the parallel capacitance can be tuned by the width and length of the defect-free part of the DGS-IMSL. Due to this increased series inductance and parallel capacitance, the DGS-IMSL behaves like a low-pass filter. On top of minimizing the reflection of the delay line, the operating frequency of the DGS-IMSL with respect to the cut-off must be chosen carefully to allow high FoM (low loss) and low NL (high compactness) while restricting nonlinearities[23]. Furthermore, a wide microstrip line with high characteristic impedance is desired in order to reduce the surface current density and therefore limit conductor loss. Dimensions can be found in Supplementary Fig. 4.

Concerning the results in Table 3, i.e., the losses of the UC, two simulations have been executed, namely a simulation of the whole UC and a simulation of the tunable delay line (both simulations are carried out

without the bias lines). The loss is evaluated at a frequency of 60 GHz with a LC relative permittivity of $\varepsilon_r = 3$ and $\tan \delta = 0.009$. The total loss of the UC was evaluated by its radiation efficiency, i.e., the ratio of gain to directivity, which corresponds to $-3.3$ dB. Since the wave has to travel twice through the UC, the total loss corresponds to $-6.6$ dB. By making use of the evaluated losses of the separate delay line simulation, the losses can be divided into losses of the delay line and losses of the radiating element. The subdivision into conductor and dielectric losses is calculated proportionally by the evaluated power lost in each material.

## Fabrication

All samples are fabricated with a standard lithographic procedure[33]. First, thin seed layers of chromium (20 nm) and gold (50 nm) are thermally evaporated on the glass samples. After structuring the surface with photoresist and a film mask (e.g., ground plane, patch, microstrip lines), the gold layer is electroplated to the desired thickness. Following the wet etching of the seed layers, SU-8 spacers are applied to the surface of one sample. Subsequently, the samples are diced to the desired size and the surface is coated with the alignment layer. After structuring the alignment layer with a velvet cloth, the samples can be bonded together. As the last steps, the RIS is filled with LC in a vacuum chamber and sealed on its edges with glue.

## Voltage optimization

A Nelder-Simplex optimization algorithm has been performed in order to adjust the voltages of the individual columns of the RIS. For this purpose, voltages were optimized towards providing maximum received power. The applied bias voltages can be found in Supplementary Tables 1 and 2. It must be noted that the applied voltages resulting from the delay line characterization in Fig. 3c already led to viable results.

## Data availability

The data that support the findings of this study are available from the corresponding author upon reasonable request.

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

## Acknowledgements
This work was funded by the Deutsche Forschungsgemeinschaft (DFG, German Research Foundation) - Project-ID 287022738 - TRR 196 MARIE within project C09. In addition, thanks goes to Merck KGaA, Darmstadt, Germany for providing the liquid crystal mixture.

## Author contributions
R.N. and A.J. conceived the experiments and design. R.N. performed the unit cell simulations. R.N. performed the layout design and fabrication. R.N. conducted the measurements and analyzed results. A.J. supervised the experiments. M.Sp. and M.Sch. contributed with their expertise in regular discussions. R.N. and A.J. wrote the manuscript and all authors reviewed it.

## Funding

## Competing interests
The authors declare no competing interests.
