## [Peer Review File · Communications Engineering]

Reviewers' comments:

Reviewer #1 (Remarks to the Author):

In this paper, a new architecture for reconfigurable intelligent surfaces based on defected LC delay lines, where the phase shifting for the backscattered signals is achieved by tunable delay lines for each antenna element, is presented. This approach is promising as it enables the design of LC-RIS with the potential of optimizing towards the bandwidth, loss, and response time simultaneously. A prototype is realized which exhibits 72ms response time, insertion losses of 6.6 dB and a 10.9% bandwidth.

The work is timely and interesting. It is one of the few works dealing with hardware realizations of RIS. The proposed implementation offers a promising alternative for the realization of RISs for future mm-Wave and THz communication and sensing systems.

Reviewer #2 (Remarks to the Author):

The authors of this work have proposed a novel architecture for LC-based RIS with compact delay lines that offer 360-degree tunability, enabling RISs with fast response time, wide bandwidth, and low loss. An RIS with a thin 4.6 μm LC layer was fabricated. The obtained module exhibits a response time of 72 ms. Implementation and measurement results are proposed. Implementation environment with equipment snapshots provided. It is clear that when compared to simulations and theoretical analysis, implementation results provide a true picture of the technology. The reviewer's comments are listed below.

Comment 1:

It is unclear how the LC delay line differs from conventional LC with regards to the limitations enumerated by the authors. These limitations are bandwidth optimization, response time, and loss. The substance refractive index provides a smooth control of the RIS elements in conventional LC. It will be beneficial to compare the LC delay line and conventional LC on the control aspect as well.

Comment 2:

It is not clear how the authors tuned the proposed LC-delay line. It will benefit the reader to know more about this operation.

Comment 3:

With regard to Fig. 1: With 72 ms response time, do the authors think the proposed design can be used

in the up coming 6G networks? If not, what can be done to shorten this time?

Comment 4:

The authors omitted a number of works that focused on LC-based RIS technology in optical wireless communications systems. Some of these works suggested the challenges enumerated and treated in this paper. The tuning time challenge has been mentioned in [R1] and [R2], while beam steering capability has been discussed in [R3], and Indium Tin Oxide was proposed in [R4]. With regard to the application of the LC-based RIS depicted in Fig. 1, this reviewer thinks the works below should be acknowledged.

[R1] A. R. Ndjiongue, T. M. N. Ngatched, O. A. Dobre and H. Haas, "Design of a Power Amplifying-RIS for Free-Space Optical Communication Systems," in *IEEE Wireless Communications*, vol. 28, no. 6, pp. 152-159, December 2021, doi: 10.1109/MWC.001.2100232.

[R2] A. R. Ndjiongue, T. M. N. Ngatched, O. A. Dobre and H. Haas, "Re-Configurable Intelligent Surface-Based VLC Receivers Using Tunable Liquid-Crystals: The Concept," in *Journal of Lightwave Technology*, vol. 39, no. 10, pp. 3193-3200, 15 May 2021, doi: 10.1109/JLT.2021.3059599.

[R3] A. R. Ndjiongue, T. M. N. Ngatched, O. A. Dobre and H. Haas, "Digital RIS (DRIS): The Future of Digital Beam Management in RIS-Assisted OWC Systems," in *Journal of Lightwave Technology*, vol. 40, no. 16, pp. 5597-5604, 15 Aug. 2022, doi: 10.1109/JLT.2022.3176762.

[R4] A. R. Ndjiongue, T. M. N. Ngatched, O. A. Dobre and H. Haas, "Toward the Use of Re-configurable Intelligent Surfaces in VLC Systems: Beam Steering," in *IEEE Wireless Communications*, vol. 28, no. 3, pp. 156-162, June 2021, doi: 10.1109/MWC.001.2000365.

Comment 5:

What is the authors' take on the limitations enumerated in Section 2.4.1?

Comment 6:

As shown in Fig. 3 (c), a bias voltage controls the phase. Apart from the physical structure, what is the main difference with the conventional LC-based RIS?

Comment 7:

Is there any technical reason for adopting a triangular pattern? If yes, how does this compare to any other pattern that could be used? Circular, rectangular?

Comment 8:

The architecture proposed in Fig. 2 (d) and (e), and Fig. 3 (a) and (h) contradict the triangular pattern announced earlier. Or maybe this reviewer is missing something.

Comment 9:

This reviewer does not understand how Table 3 parameters are obtained. The proposed method and measurement do not clarify this. Please elaborate more.

Comment 10:

The writing could be improved:

The last sentence before Section 4 (Methods) reads poorly. A comma is missing before an "and".

Equations (1) and (2) should contain a punctuation.

Some abbreviations are unclear.. CMOS for example.

A dot is missing after the key words.

Most definitions contain a capital letter in the first letter. This reviewer does not understand why. Examples are: Liquid Crystal, Millimeter Wave, Unit Cell (UC), Radio Frequency (RF), etc.

"...year, we [20] and Kim et al. [21] proposed a compact, low loss..." This sentence does not read well.

"We [20]. ...

For that purpose, a LC-RIS with thin ----- an RIS?

Peer Review File

Reviewers' comments:

Reviewer #1 (Remarks to the Author):

In this paper, a new architecture for reconfigurable intelligent surfaces based on defected LC delay lines, where the phase shifting for the backscattered signals is achieved by tunable delay lines for each antenna element, is presented. This approach is promising as it enables the design of LC-RIS with the potential of optimizing towards the bandwidth, loss, and response time simultaneously. A prototype is realized which exhibits 72ms response time, insertion losses of 6.6 dB and a 10.9% bandwidth.

The work is timely and interesting. It is one of the few works dealing with hardware realizations of RIS. The proposed implementation offers a promising alternative for the realization of RISs for future mm-Wave and THz communication and sensing systems.

Reviewer #2 (Remarks to the Author):

The authors of this work have proposed a novel architecture for LC-based RIS with compact delay lines that offer 360-degree tunability, enabling RISs with fast response time, wide bandwidth, and low loss. An RIS with a thin 4.6 μm LC layer was fabricated. The obtained module exhibits a response time of 72 ms. Implementation and measurement results are proposed. Implementation environment with equipment snapshots provided. It is clear that when compared to simulations and theoretical analysis, implementation results provide a true picture of the technology. The reviewer's comments are listed below.

Comment 1:

It is unclear how the LC delay line differs from conventional LC with regards to the limitations enumerated by the authors. These limitations are bandwidth optimization, response time, and loss. The substance refractive index provides a smooth control of the RIS elements in conventional LC. It will be beneficial to compare the LC delay line and conventional LC on the control aspect as well.

Comment 2:

It is not clear how the authors tuned the proposed LC-delay line. It will benefit the reader to know more about this operation.

Comment 3:

With regard to Fig. 1: With 72 ms response time, do the authors think the proposed design can be used in the up coming 6G networks? If not, what can be done to shorten this time?

Comment 4:

The authors omitted a number of works that focused on LC-based RIS technology in optical wireless communications systems. Some of these works suggested the challenges

enumerated and treated in this paper. The tuning time challenge has been mentioned in [R1] and [R2], while beam steering capability has been discussed in [R3], and Indium Tin Oxide was proposed in [R4]. With regard to the application of the LC-based RIS depicted in Fig. 1, this reviewer thinks the works below should be acknowledged.

[R1] A. R. Ndjiongue, T. M. N. Ngatched, O. A. Dobre and H. Haas, "Design of a Power Amplifying-RIS for Free-Space Optical Communication Systems," in *IEEE Wireless Communications*, vol. 28, no. 6, pp. 152-159, December 2021, doi: 10.1109/MWC.001.2100232.

[R2] A. R. Ndjiongue, T. M. N. Ngatched, O. A. Dobre and H. Haas, "Re-Configurable Intelligent Surface-Based VLC Receivers Using Tunable Liquid-Crystals: The Concept," in *Journal of Lightwave Technology*, vol. 39, no. 10, pp. 3193-3200, 15 May 15, 2021, doi: 10.1109/JLT.2021.3059599.

[R3] A. R. Ndjiongue, T. M. N. Ngatched, O. A. Dobre and H. Haas, "Digital RIS (DRIS): The Future of Digital Beam Management in RIS-Assisted OWC Systems," in *Journal of Lightwave Technology*, vol. 40, no. 16, pp. 5597-5604, 15 Aug. 15, 2022, doi: 10.1109/JLT.2022.3176762.

[R4] A. R. Ndjiongue, T. M. N. Ngatched, O. A. Dobre and H. Haas, "Toward the Use of Re-configurable Intelligent Surfaces in VLC Systems: Beam Steering," in *IEEE Wireless Communications*, vol. 28, no. 3, pp. 156-162, June 2021, doi: 10.1109/MWC.001.2000365.

Comment 5:

What is the authors' take on the limitations enumerated in Section 2.4.1?

Comment 6:

As shown in Fig. 3 (c), a bias voltage controls the phase. Apart from the physical structure, what is the main difference with the conventional LC-based RIS?

Comment 7:

Is there any technical reason for adopting a triangular pattern? If yes, how does this compare to any other pattern that could be used? Circular, rectangular?

Comment 8:

The architecture proposed in Fig. 2 (d) and (e), and Fig. 3 (a) and (h) contradict the triangular pattern announced earlier. Or maybe this reviewer is missing something.

Comment 9:

This reviewer does not understand how Table 3 parameters are obtained. The proposed method and measurement do not clarify this. Please elaborate more.

Comment 10:

The writing could be improved:

The last sentence before Section 4 (Methods) reads poorly. A comma is missing before an "and".

Equations (1) and (2) should contain a punctuation.

Some abbreviations are unclear.. CMOS for example.

A dot is missing after the key words.

Most definitions contain a capital letter in the first letter. This reviewer does not understand why. Examples are: Liquid Crystal, Millimeter Wave, Unit Cell (UC), Radio Frequency (RF), etc.

"...year, we [20] and Kim et al. [21] proposed a compact, low loss..." This sentence does not read well. "We [20]. ...

For that purpose, a LC-RIS with thin ----- an RIS?

Reviewer #1 (Remarks to the Author):

In this paper, a new architecture for reconfigurable intelligent surfaces based on defected LC delay lines, where the phase shifting for the backscattered signals is achieved by tunable delay lines for each antenna element, is presented. This approach is promising as it enables the design of LC-RIS with the potential of optimizing towards the bandwidth, loss, and response time simultaneously. A prototype is realized which exhibits 72ms response time, insertion losses of 6.6 dB and a 10.9% bandwidth.

The work is timely and interesting. It is one of the few works dealing with hardware realizations of RIS. The proposed implementation offers a promising alternative for the realization of RISs for future mm-Wave and THz communication and sensing systems.

We appreciate the reviewer's positive evaluation and acknowledgment of the significance of our manuscript.

Reviewer #2 (Remarks to the Author):

The authors of this work have proposed a novel architecture for LC-based RIS with compact delay lines that offer 360-degree tunability, enabling RISs with fast response time, wide bandwidth, and low loss. An RIS with a thin 4.6 μm LC layer was fabricated. The obtained module exhibits a response time of 72 ms. Implementation and measurement results are proposed. Implementation environment with equipment snapshots provided. It is clear that when compared to simulations and theoretical analysis, implementation results provide a true picture of the technology. The reviewer's comments are listed below.

Comment 1:

It is unclear how the LC delay line differs from conventional LC with regards to the limitations enumerated by the authors. These limitations are bandwidth optimization, response time, and loss. The substance refractive index provides a smooth control of the RIS elements in conventional LC. It will be beneficial to compare the LC delay line and conventional LC on the control aspect as well.

The reviewer's input is greatly appreciated. In this comment, the reviewer asks for the explanation of two things:

- a) The differentiation of the DLA and the RA with respect to the limitations regarding bandwidth, response time and loss
- b) A more detailed explanation of the control aspect (biasing) of the LC-RIS

a) Regarding the differentiation of the DLA and RA with respect to limitations in bandwidth, response time and loss, the following points are stated in the manuscript:

The tradeoff with the RA is explained in lines 92 to 101:

“There already exist promising realizations of LC-reflectarrays and LC-RISs based on RA showing the potential for LC-RISs to operate even at THz-frequencies. For example, [16] and [17] operating at frequencies around 100 GHz and 400 GHz, respectively. However, one crucial limitation of the RA lies in thick LC layers (10s to

100s of μm) needed to achieve sufficient bandwidths, which lead to slow response times (10s of seconds [18]). Lately, efforts emerged to build RA implementations towards millisecond response time [19], but at the expense of considerably decreased LC tunability and increased losses. Hence, with the LC-RA method, trade-offs between bandwidth, loss, and response time are inevitable. In other words, the optimization of two aspects concurrently requires making concessions in the remaining aspect.”

In other words: Since in RA applications, the LC serves as both the radiating and the phase shifting substrate, thick LC layers are required to enable efficient performance (i.e. electromagnetic coupling) between the antenna and free space over large bandwidths. However, the thick LC layers lead to slow response. The slow response times are tried to be overcome by special LC mixtures, which, however, lead to higher losses.

As a result, it is not yet possible to overcome the tradeoff between the three mentioned aspects with the RA, although the LC is continuously tunable as the reviewer mentioned.

With the DLA, the principle to overcome the tradeoff is described in lines 134 to 147:

“The main difference of the DLA in comparison to the RA is the separation of the phase shifting and the radiating layer, which provides three benefits:

- 1) As the phase shifting mechanism does not originate from a resonance effect, amplitude variations are minimized and the bandwidth is mainly dictated by the radiating element.*
- 2) The radiating layer can be realized with an electrically thick dielectric material to maximize bandwidth, while the LC layer (t_{LC}) can be kept thin. A thin t_{LC} is decisive, as the switch on response time τ_{on} (from $\epsilon_{r,\perp}$ to $\epsilon_{r,\parallel}$) and switch off response time τ_{off} (from $\epsilon_{r,\parallel}$ to $\epsilon_{r,\perp}$) increase quadratically with the LC thickness, t_{LC} [22].*
- 3) The DLA provides high flexibility, as the tunable delay line can be engineered towards application-specific requirements with relaxed constraints of the selected radiating layer topology.*

With the aforementioned benefits, it is possible to optimize a RIS towards low loss, wide bandwidth and fast response time simultaneously.”

In other words: The radiating layer and the phase shifting layer are separated in the DLA. The radiating layer can be an electrically thick substrate (e.g. glass) to enable large bandwidths while the phase shifting layer is a tunable delay line which can be realized with a thin LC layer to provide fast response times. With the design of a suitable tunable delay line, the loss can also be kept low, such that the tradeoff between loss, bandwidth and response time can be overcome.

In essence, our opinion is that the manuscript provides sufficient information on why the DLA is superior to the RA in terms of optimization of loss, bandwidth and response time simultaneously. We trust the reviewer finds this perspective well-founded within the manuscript.

b) Regarding the control aspect:

Here, the authors agree with the reviewer that it will be beneficial for the reader if some details are added. The principle aspect of biasing (controlling) the LC is the same for the RA and the DLA. For both, the LC is placed between two metal electrodes between which a bias voltage is applied so that the LC can be continuously tuned. Notably, DLA's reliance on thinner LC layers allows for lower bias voltages, presenting an advantage over RA. Additionally, if the DLA supports dual linear polarizations, it facilitates independent polarization control for a single radiating element, unlike the RA requiring two radiating elements for this purpose.

These points have been added to the manuscript in Lines 192 to 195 and 257 to 263.

Comment 2:

It is not clear how the authors tuned the proposed LC-delay line. It will benefit the reader to know more about this operation.

In this comment, the reviewer is referring to the tuning of the applied LC delay line and requests more details.

Thank you for your feedback. The authors aimed to address both aspects concerning the "tuning" of the proposed LC-delay line, i.e. its initial optimization and the biasing in measurements. To clarify these operations, significant information has been added in the Methods section. Specifically, details on the optimization process for the delay line have been elaborated (lines 404 to 415). Additionally, further insights into the biasing procedure of the delay line have been provided (lines 364 to 372). These additions have been made to enhance the comprehensiveness of the manuscript, providing a more thorough understanding of the tuning operations involved.

Comment 3:

With regard to Fig. 1: With 72 ms response time, do the authors think the proposed design can be used in the up coming 6G networks? If not, what can be done to shorten this time?

The reviewer's question relates to the applicability of the proposed design in upcoming 6G networks, especially regarding the comparably slow response time of LC-based devices in general.

To give some background on the envisioned applications, section 2.4.3 "Application Perspective" has been added, evaluating for which applications the proposed LC-RIS architecture is envisioned (lines 320 to 343).

Regarding the question of what can be done to shorten the response time, suitable methods are already presented in the manuscript in Section 2.4.2 under the point "Response Time" (Lines 302 to 311). The methods include overdriving and undershooting techniques and thinner LC layers. Another method has been added, which is the dilution of polymerizable monomers in the LC-mixtures, which create strong anchoring forces to reduce the response time. However, the addition comes with the disadvantage of higher losses and higher required bias voltages.

Comment 4:

The authors omitted a number of works that focused on LC-based RIS technology in optical wireless communications systems. Some of these works suggested the challenges enumerated and treated in this paper. The tuning time challenge has been mentioned in [R1] and [R2], while beam steering capability has been discussed in [R3], and Indium Tin Oxide was proposed in [R4]. With regard to the application of the LC-based RIS depicted in Fig. 1, this reviewer thinks the works below should be acknowledged.

[R1] A. R. Ndjiongue, T. M. N. Ngatched, O. A. Dobre and H. Haas, "Design of a Power Amplifying-RIS for Free-Space Optical Communication Systems," in IEEE Wireless Communications, vol. 28, no. 6, pp. 152-159, December 2021, doi: 10.1109/MWC.001.2100232.

[R2] A. R. Ndjiongue, T. M. N. Ngatched, O. A. Dobre and H. Haas, "Re-Configurable Intelligent Surface-Based VLC Receivers Using Tunable Liquid-Crystals: The Concept," in Journal of Lightwave Technology, vol. 39, no. 10, pp. 3193-3200, 15 May 15, 2021, doi: 10.1109/JLT.2021.3059599.

[R3] A. R. Ndjiongue, T. M. N. Ngatched, O. A. Dobre and H. Haas, "Digital RIS (DRIS): The Future of Digital Beam Management in RIS-Assisted OWC Systems," in Journal of Lightwave Technology, vol. 40, no. 16, pp. 5597-5604, 15 Aug. 15, 2022, doi: 10.1109/JLT.2022.3176762.

[R4] A. R. Ndjiongue, T. M. N. Ngatched, O. A. Dobre and H. Haas, "Toward the Use of Re-configurable Intelligent Surfaces in VLC Systems: Beam Steering," in IEEE Wireless Communications, vol. 28, no. 3, pp. 156-162, June 2021, doi: 10.1109/MWC.001.2000365.

Thank you for providing these valuable references. In response to the mentioned omissions, Reference [R4] has been included in the manuscript (lines 85 and 86) to expand the readers' understanding that LC-RIS concepts are also being envisioned for application in the optical domain.

Comment 5

What is the authors' take on the limitations enumerated in Section 2.4.1?

The authors tackled some of the limitations in the manuscript. An additional point was added to the manuscript:

The described limitations are:

- Biasing

This is a challenge already solved in LCDs. The techniques already applied in LCDs, i.e. Active Matrix Wise Biasing, should be applied for the DLA (see lines 250 to 254).

- Polarization

For dual linear polarization, more compact delay line topologies are required to fit in the unit cell. For this purpose, for example bandpass-based approaches can be used as presented in the reference [24] (see lines 255 to 263).

- Temperature Dependence

Temperature dependence can be solved with temperature sensing systems. This information is added to the manuscript (see lines 268 to 270).

- Response Time

Reduction of response time is presented in 2.4.2 in “Response Time” and includes overdriving and undershooting, thinner LC layers and the addition of polymerizable monomers to the LC mixture (see lines 307 to 311).

Comment 6:

As shown in Fig. 3 (c), a bias voltage controls the phase. Apart from the physical structure, what is the main difference with the conventional LC-based RIS?

The reviewer noticed how the bias voltage is controlled in Figure 3c) and asked about the main difference between the DLA and RA in terms of biasing.

In response, there is no significant difference in the principle of controlling the bias voltage between DLA and RA approaches besides the physical structure. However, due to the reduced layer thickness to conventional RA approaches, lower voltages are required for DLA approaches. This information has been added to the manuscript (see lines 192 to 195).

Comment 7:

Is there any technical reason for adopting a triangular pattern? If yes, how does this compare to any other pattern that could be used? Circular, rectangular?

The triangular pattern provides a similar performance to a conventional rectangular array. In case of this contribution, however, it provides additional space for the delay line due to the operation with single linear polarization, relaxing the requirements for the delay line in terms of compactness. This is stated in lines 184 to 188 in the manuscript.

If a more compact delay line topology is applied, a rectangular array could also be adopted.

Comment 8:

The architecture proposed in Fig. 2 (d) and (e), and Fig. 3 (a) and (h) contradict the triangular pattern announced earlier. Or maybe this reviewer is missing something.

Figure 3 a) doesn't show any grid, it is just a picture of a separate phase shifter sample. The grid of the RIS in Fig. 2 d) 2 e) and 3 h) show a triangular pattern.

Maybe Fig. 2c), which shows the principle DLA architecture with a rectangular grid in the initially submitted manuscript, was misleading. It has been changed to a triangular arranged grid to avoid confusion. The authors express their appreciation to the reviewer for bringing attention to this potentially misleading depiction.

Comment 9:

This reviewer does not understand how Table 3 parameters are obtained. The proposed method and measurement do not clarify this. Please elaborate more.

The reviewer raised a valid point about the lack of clarity on how the parameters in Table 3 were obtained. For that purpose, an explanation of the procedure has been added to the manuscript in the Methods section (Lines 416 to 426). We appreciate the reviewer for identifying this lack of information, as their input has contributed to enhancing the clarity and completeness of our work.

Comment 10:

The writing could be improved:

The last sentence before Section 4 (Methods) reads poorly. A comma is missing before an "and".

Equations (1) and (2) should contain a punctuation.

Some abbreviations are unclear.. CMOS for example.

A dot is missing after the key words.

Most definitions contain a capital letter in the first letter. This reviewer does not understand why. Examples are: Liquid Crystal, Millimeter Wave, Unit Cell (UC), Radio Frequency (RF), etc.

"...year, we [20] and Kim et al. [21] proposed a compact, low loss..." This sentence does not read well. "We [20]. ..."

For that purpose, a LC-RIS with thin ----- an RIS?

The authors would like to thank the reviewer for improving the writing in the manuscript. The mentioned writing issues are all addressed in the new version.

REVIEWERS' COMMENTS:

Reviewer #2 (Remarks to the Author):

The authors addressed my concerns. However, this reviewer believes they have not given enough recognitions to previous work on systems assisted by liquid crystal-based RIS. With regard to this question, the authors wrote:

"Thank you for providing these valuable references. In response to the mentioned omissions, Reference [R4] has been included in the manuscript (lines 85 and 86) to expand the readers' understanding that LC-RIS concepts are also being envisioned for application in the optical domain."

This response gives the impression that LC-RIS is proposed for optical systems the way it is proposed for RF. This is not the case because LCs are mostly used in the optical domain and there has been lots of work done in that domain. LCs are used in several applications using optical signals. I think the authors should check the literature and give more recognition to previous works on systems assisted by LC-based RISs.

Peer Review File

Reviewers' comments:

Reviewer #2 (Remarks to the Author):

The authors addressed my concerns. However, this reviewer believes they have not given enough recognitions to previous work on systems assisted by liquid crystal-based RIS. With regard to this question, the authors wrote:

"Thank you for providing these valuable references. In response to the mentioned omissions, Reference [R4] has been included in the manuscript (lines 85 and 86) to expand the readers' understanding that LC-RIS concepts are also being envisioned for application in the optical domain."

This response gives the impression that LC-RIS is proposed for optical systems the way it is proposed for RF. This is not the case because LCs are mostly used in the optical domain and there has been lots of work done in that domain. LCs are used in several applications using optical signals. I think the authors should check the literature and give more recognition to previous works on systems assisted by LC-based RISs.

In order to give more recognition to previous works on liquid crystal-based RIS in visible light communications, the authors have added references [17] and [18]. Furthermore, the extensive existing knowledge in LC mixtures and devices in the optical domain has been noted.